# Health condition at first fit note and number of fit notes: a longitudinal study of primary care records in south London

Sarah Dorrington [1,2] Ewan Carr [1] C Polling [1,2] Sharon Stevelink,[1] Mark Ashworth [3] Emmert Roberts [2,4] Matthew Broadbent,[2] Stephani Hatch,[1] Ira Madan,[5] Matthew Hotopf[1,2]

¹Department of Psychological Medicine, King's College London Institute of Psychiatry Psychology and Neuroscience, London, UK
²NIHR Maudsley Biomedical Research Centre, South London and Maudsley Mental Health NHS Trust, London, UK
³School of Population Health and Environmental Sciences, King's College London, London, UK
⁴National Addiction Centre, King's College London Institute of Psychiatry Psychology and Neuroscience, London, UK
⁵Department of Occupational Health, Guy's and St Thomas' Hospitals NHS Trust, London, UK

**Correspondence to**
Sarah Dorrington;
sarah.dorrington@kcl.ac.uk

## ABSTRACT

**Objectives** The fit note replaced the sick note in the UK in 2010, with the aim of improving support for patients requiring sickness absence, yet there has been very little research into fit note use. This study aims to describe number of fit notes by condition, to improve our understanding of patterns of fit note use in primary care. Previous fit note research has relied on extracting diagnoses directly from fit notes, rather than extracting information from clinical records. In this paper, we extract information from clinical records to explore demographic factors and conditions associated with number of fit notes issued.

**Design** This is a longitudinal study of clinical data. We analysed individual-level anonymised data from general practitioner consultations, including demographic information and condition recorded at first fit note. The latter encompassed diagnoses, individual symptoms and psychosocial issues.

**Setting** A database called Lambeth DataNet, containing electronic clinical records on 326 415 adults (ages 16–60) from all 45 general practices within the London Borough of Lambeth from 1 January 2014 to 30 April 2017.

**Participants** Our analytical sample contained 40 698 people with a condition recorded at first fit note.

**Primary outcome measure** Predicted number of fit notes in the period January 2014–April 2017

**Results** Of all studied diagnostic groups, mental illness had the highest predicted number of fit notes (n=3.3; 95% CI: 3.1 to 3.4) after controlling for demographic factors and long-term conditions. The highest predicted number of fit notes for any condition subgroup was among patients presenting for drug and/or alcohol misuse (n=4.5; 95% CI: 4.1 to 4.8).

**Conclusions** For the first time, we show drug and/or alcohol misuse at first fit note are associated with the highest number of fit notes. Research is needed to understand the trajectories of individuals at highest risk of long-term sickness absence, in particular, people presenting with drug and/or alcohol misuse.

## Strengths and limitations of this study

► This is the largest study to analyse fit note use and clinical information from general practitioner (GP) records at point of first fit note, using individual-level clinical data; a major strength of using clinical records data is that they contain more detail on condition type than information reported on the fit note which previous research has relied on.

► The study was conducted in a single geographical area for which we had GP data; this had the benefit of providing information on virtually all individuals seeking help within that population, but it had the disadvantage of a possible loss of generalisability, and our findings should be tested in other areas.

► We do not have access to the information written on the fit note to compare it with the information recorded in clinical notes, and we do not know the length of time that each fit note was prescribed for.

► Each of the condition groups and subgroups encompasses a very wide variety of disorders.

► There is additional information that would have been extremely useful, such as nature of employment, educational level, occupational status or benefit use, which were not available in this data set.

usually expressed in financial terms, with an estimated £15bn lost to the economy in the year 2017/18[2]. In 2008, a government report suggested that the process of sickness certification by general practitioners (GPs) was a contributor to the problem of long-term sickness absence, and recommended the introduction of a new form of sickness certification: the fit note.[2]

The sick note, used to certify episodes of sickness absence of over 7 days, was replaced by the fit note in 2010.[3] The fit note was designed to reduce long-term sickness absence by replacing the sick note's binary 'fit' versus 'not fit to work', with the addition of a third option, 'maybe fit',[2] and encouraging GPs to certify shorter periods of sickness absence, with more frequent reviews. 'Not fit

## INTRODUCTION

In the UK, long-term sickness absence is a major driver of inequalities, leading to social exclusion and adverse health outcomes.[1] The impact of long-term sickness absence is

for work' and the new 'maybe fit' option both provide evidence of sickness to an employer and can be used in the administration of sickness benefits. The fit note also includes space for GPs to recommend work adjustments for employees, to encourage employers to adapt to the health needs of workers.[4] The digitisation of the fit note in 2012 created the opportunity for researchers and policy-makers to describe patterns of fit note use in the UK population.[5–7]

A single fit note following an injury or illness is a common occurrence. A study found that 60% of all fit notes issued in the UK from 2011 to 2013 were single fit notes.[8] In contrast, repeated fit notes are rarer and suggest a more serious injury or chronic illness, or other reasons for not recovering, such as multimorbidity, illness complications or barriers to return to work. Identifying conditions and demographic groups associated with high fit note use enables us to define groups with high levels of need and to inform future interventions.

Mental disorders are the single most common reason for a fit note, making up over 30% of fit notes nationally (NHS Digital 2014–2017).[5] Research to date has found multiple fit note use to be associated with mental illness and stress-related problems, higher levels of deprivation, minority ethnic groups and female gender.[9–11] In the majority of fit note research, the reason for sickness absence is taken directly from the fit note, where information is written for the worker, employer and/or the Department of Work and Pensions.[12 13] There are limitations to this approach. First, half of all fit notes do not provide adequate information for the reason for sickness absence to be coded,[14] and second, information on the fit note may differ from information in the clinical notes, for reasons such as individual confidentiality or clinicians' preference.[13 15 16]

International research suggests that a large proportion of workers presenting for sickness certification have multimorbidity;[17 18] yet, the extent of multimorbidity is rarely captured on the fit note itself, even though there is no limit to the number of conditions that can be recorded. Even when information about multimorbidity is provided on the fit note, it is often excluded from analysis.[8 19]

In this paper, we overcome some of the limitations of relying on information provided on the fit note by extracting conditions recorded in clinical records at the point of first fit note receipt from Read codes recorded by GPs. Conditions include diagnoses, individual symptoms, psychosocial issues and codes related to the healthcare provided. We explore the relationship between variables known to be associated with fit note receipt and number of fit notes, anticipating that the number of fit notes issued in our sample, between January 2014 and April 2017, would be higher among individuals with mental disorders, those with multimorbidity, women, those living in areas of greater deprivation, older age groups and minority ethnic groups. We will test for a multiplier effect of mental health and physical comorbidity on number of fit notes. We expect demographic associations to be

partially explained by long-term conditions and that individuals with infection, injury and obstetric conditions will tend to have a fewer fit notes.

In summary, the first aim of this paper is to describe the predicted number of fit notes by condition at first fit note extracted from clinical notes. The second objective is to describe demographic variation in predicted number of fit notes. Finally, we will explore the interaction effects of comorbid conditions at first fit note on predicted number of fit notes.

## METHODS

### Data

Prospective longitudinal data for 326 415 patients were extracted from electronic clinical records using a database of primary care providers in the London Borough of Lambeth, Lambeth DataNet (LDN).[11] The London Borough of Lambeth, home to a population of just over 318 000, is the 22nd most deprived local authority in England[20] and has the fourth highest level of income inequality of any borough in London.[21] Lambeth contains a young and ethnically diverse population, including large Portuguese, South American and Black populations.[22] Around 150 different languages are spoken and around 17 000 people (6% of the population) speak a main language other than English.[20] Thirty-eight percent of the population was born outside of the UK.[20] The number of individuals registered on LDN at any one time (n=405 000) exceeds the total Lambeth population recorded in the Census, due to the cross-boundary registration of individuals not living in Lambeth, GP list inflation[23] and the temporary residence of students in the Lambeth area from surrounding universities. We analysed records covering a period of 40 months from 1 January 2014 to 30 April 2017. Data were extracted in May 2017 from the primary care clinical record system, EMIS Web. We restricted the sample to adults age 16–60 years to exclude exits from the workforce due to reaching statutory pension age (ages 60–65 years), which was not measured in our sample.

### Demographic variables

Information on individuals' first recorded sex, age at the beginning of window, ethnicity and lower super output area (LSOA) of residence (an area covering an average population of 1722)[24] were extracted from the GP records. Ethnicity was coded using seven subcategories: Asian, Black African, Black Caribbean, Black Other, Mixed, Other and White. This uses the five broad groups used in the 2011 census, with the Black group split into subgroups, following previous research within South London suggesting differences in mental health outcomes between Black Caribbean and Black African groups.[25 26] Deprivation was measured using the Index of Multiple Deprivation (IMD) 2015 rank of their LSOA of residence. IMD scores were divided into quintiles, based on deprivation scores in Lambeth.[27]

## Fit note

Date of first fit note was derived for each participant based on digital fit notes issued between 1 January 2014 and 30 April 2017. We excluded fit notes issued before this period because although digital fit notes were introduced in 2012, they were not in widespread use until 2014. Due to the lack of information on prior sickness certification (before January 2014), we could not determine whether a 'first fit note' during the study period represented the individual's first ever sickness certification. Number of fit notes are measured as the total number of fit notes received in the study window.

## Presenting condition at first fit note

There were a wide variety of conditions (recorded using Read codes) in the electronic GP record when a first fit note was issued. These were split into two broad categories of 'mental health' and 'non-mental health', containing 11 broad groups comprising 37 subgroups. The creation of 11 groups was based on a pragmatic mix of the size of the groups, the way diagnoses are grouped in the International Classification of Diseases (ICD-10) and in past fit note research.[19 28 29] The three mental health groups were mental disorder, stress and specific external stressors (which includes bereavement and relationship breakdown) and the eight non-mental health groups were musculoskeletal, infection, surgery, obstetrics, injury, physical health, fatigue and a group for 'other' conditions that could not be easily categorised into one of the other groups (eg, administrative codes). The broad 'physical health' group was created for individual symptoms that could not be attributed to a single system or group of diagnoses. We defined 37 diagnostic subgroups (online supplemental figure 1 and table 1). Where a condition fit into more than one diagnostic group, we used a hierarchy to allocate codes ensuring each condition was allocated to a single diagnostic subgroup (see online supplemental figure 1). For patients with more than one condition at first fit note (14% of people), one condition was selected at random as the primary condition, additional conditions were taken into account in the analysis of multiple symptoms.

## Long-term conditions and pain

Presence of long-term conditions was assessed in addition to condition at first fit note, using the Quality and Outcomes Framework (QOF), an annual reward and incentive programme for all GP practices in England. QOF measures have been shown to underestimate prevalence of disorders in the general population due to the reliance on the GP for a diagnosis.[30] QOF diagnosis is therefore a specific, but not sensitive measure[31] with a low false-positive but high false-negative rates. In this paper, 'long-term condition' comprises 15 QOF[32] conditions: depression, epilepsy, diabetes mellitus, coronary heart disease, chronic obstructive pulmonary disorder, cancer (non-specified), atrial fibrillation, heart failure, stroke, rheumatoid arthritis, obesity, severe mental illness (SMI;

schizophrenia, bipolar disorder), learning disability, hypertension and asthma. QOF conditions excluded from our analysis were smoking (not a long-term condition), palliative care, osteoporosis, dementia and chronic kidney disease (due to the small numbers with these diagnosis in the age group under study). In addition, chronic pain[33] was derived based on receipt (yes/no) of any prescriptions listed in British National Formulary medication chapters 4.7.2 or 4.7.3 (opioid analgesics and neuropathic pain medication) with repeat, repeat dispensed or automatic issue type.

## Statistical analysis

We undertook a descriptive analysis of demographic variables (age, gender, ethnicity and deprivation), long-term conditions and conditions at first fit note. To assess associations between conditions and fit note counts, we used zero-truncated negative binomial regression models. A truncated model was used because all individuals in our sample had at least one fit note to ensure predictions were made on the range $[1, \infty)$ rather than $[0, \infty)$. A negative binomial model was used to account for overdispersion in the number of fit notes. To account for clustering of individuals within GP practices, standard errors were adjusted using a sandwich estimator.[34] Models are presented in terms of model predicted number of fit notes because this is the most clinically relevant outcome. The predicted number of fit notes was calculated for each group, holding other variables to their sample mean values. Regarding the difference between predicted and actual counts, these are connected but different. The actual counts are the counts observed in the raw data, without any adjustment or clustering. The predicted counts are derived from the regression model, and as such, account for other variables in the model. In model 1, we adjust for demographic variables: age, gender, ethnicity and deprivation, and in model 2, we adjust for demographic variables and additionally, the number of long-term conditions. We did a full case analysis, described in detail in a recent publication.[11] Finally, we test for an interaction between comorbid conditions and predicted number of fit notes.

## RESULTS

Our analytical sample included 40 698 patients with at least one condition code recorded at first fit note, having excluded 284 913 patients without any fit notes, and 804 people with undefined reasons for fit note use, these included administrative codes reported in the clinical record at point of first fit note. Of 40 698 patients, 86% of individuals had one condition code and 14% of people had more than one condition recorded in the clinical notes at first fit note. The number of conditions types varied by age, with 13% of 16–40-year olds having multiple condition types compared with 17% among 51–60-year olds (p<0.001). The largest group of conditions was the physical health group (31% of all conditions), followed

**Table 1** Number of fit notes received and predicted number of fit notes by sociodemographic variables

| Demographic variables | Number of fit notes received: 1 January 2014 to 30 April 2017 n (%) | | | | Predicted NFN unadjusted (95% CI) | Predicted number of fit notes (95% CI) | |
|---|---|---|---|---|---|---|---|
| | 1 | 2 | 3 | 4+ | | Model 1* | Model 2† |
| Whole population | 40 698 | | | | 2.0 (1.8 to 2.1) | 2.2 (2.1 to 2.3) | 2.4 (2.3 to 2.5) |
| Gender | | | | | | | |
| Male | 6097 (35.2) | 3014 (17.4) | 1878 (10.9) | 6312 (36.5) | 2.0 (1.8 to 2.2) | 2.2 (2.1 to 2.3) | 2.4 (2.3 to 2.6) |
| Female | 8304 (35.5) | 4231 (18.1) | 2557 (10.9) | 8305 (35.5) | 1.9 (1.8 to 2.1) | 2.2 (2.1 to 2.3) | 2.3 (2.2 to 2.4) |
| Age group | | | | | | | |
| 16–20 | 428 (46.9) | 178 (19.5) | 80 (8.8) | 227 (24.9) | 1.4 (1.2 to 1.6) | 1.4 (1.2 to 1.6) | 1.7 (1.5 to 1.9) |
| 21–25 | 2054 (49.3) | 807 (19.4) | 379 (9.1) | 926 (22.2) | 1.3 (1.2 to 1.5) | 1.4 (1.3 to 1.5) | 1.7 (1.5 to 2.8) |
| 36–30 | 2825 (46.0) | 1138 (18.5) | 650 (10.6) | 1532 (24.9) | 1.5 (1.4 to 1.6) | 1.6 (1.4 to 1.7) | 1.8 (2.7 to 2.9) |
| 31–35 | 2311 (38.7) | 1136 (19.0) | 691 (11.6) | 1839 (30.8) | 1.8 (1.7 to 2.0) | 1.9 (2.8 to 2.0) | 2.2 (2.1 to 2.3) |
| 36–40 | 1771 (34.3) | 996 (19.3) | 591 (11.5) | 1803 (34.9) | 2.1 (2.0 to 2.3) | 2.2 (2.0 to 2.3) | 2.4 (2.3 to 2.5) |
| 41–45 | 1541 (31.2) | 881 (17.8) | 532 (10.8) | 1985 (40.2) | 2.4 (2.3 to 2.6) | 2.5 (2.3 to 2.6) | 2.6 (2.4 to 2.7) |
| 46–50 | 1528 (28.1) | 880 (16.2) | 594 (10.9) | 2437 (44.8) | 2.7 (2.5 to 2.9) | 2.7 (2.6 to 2.9) | 2.7 (2.5 to 2.8) |
| 51–55 | 1226 (26.2) | 709 (15.1) | 536 (11.4) | 2218 (47.3) | 2.9 (2.7 to 3.1) | 2.9 (1.8 to 3.1) | 2.8 (2.6 to 2.9) |
| 56–60 | 717 (21.9) | 520 (15.9) | 382 (11.7) | 1650 (50.5) | 3.1 (2.9 to 3.3) | 3.2 (3.0 to 3.4) | 2.9 (2.7 to 3.0) |
| Ethnicity | | | | | | | |
| White | 8592 (39.4) | 3979 (18.2) | 2264 (10.4) | 6980 (32.0) | 1.8 (1.6 to 2.0) | 2.0 (1.9 to 2.1) | 2.2 (2.1 to 2.3) |
| Black African | 1889 (32.4) | 1090 (18.7) | 650 (11.1) | 2211 (37.9) | 2.1 (2.0 to 2.3) | 2.2 (2.1 to 2.4) | 2.4 (2.2 to 2.5) |
| Asian | 845 (35.5) | 430 (18.1) | 292 (12.3) | 815 (34.2) | 1.9 (1.7 to 2.2) | 2.2 (1.9 to 2.4) | 2.3 (2.1 to 2.5) |
| Black Caribbean | 1279 (25.4) | 809 (16.1) | 567 (11.3) | 2380 (47.3) | 2.7 (2.5 to 2.9) | 2.8 (2.7 to 2.9) | 2.9 (2.8 to 3.0) |
| Mixed | 774 (31.5) | 404 (16.5) | 300 (12.2) | 978 (39.8) | 2.2 (2.1 to 2.4) | 2.6 (2.4 to 2.8) | 2.7 (2.6 to 2.9) |
| Other | 503 (36.2) | 268 (19.3) | 167 (12.0) | 450 (32.4) | 1.9 (1.7 to 2.1) | 2.0 (1.8 to 2.2) | 2.3 (2.1 to 2.5) |
| Black Other | 519 (29.1) | 265 (14.9) | 195 (10.9) | 803 (45.1) | 2.6 (2.4 to 2.8) | 2.8 (2.6 to 3.0) | 2.9 (2.7 to 3.2) |
| Deprivation | | | | | | | |
| 1 | 2325 (39.8) | 1142 (19.5) | 626 (10.7) | 1750 (30.0) | 1.7 (1.5 to 1.8) | 1.9 (1.8 to 2.0) | 2.0 (2.0 to 2.2) |
| 2 | 2575 (38.1) | 1265 (18.1) | 705 (10.4) | 2254 (33.3) | 1.8 (1.7 to 2.0) | 2.1 (2.9 to 2.2) | 2.1 (2.1 to 2.4) |
| 3 | 2725 (37.6) | 1411 (17.1) | 823 (11.4) | 2836 (33.9) | 1.9 (1.7 to 2.1) | 2.1 (2.0 to 2.3) | 2.1 (2.1 to 2.4) |
| 4 | 3067 (32.7) | 1606 (17.8) | 1033 (11.0) | 3521 (38.5) | 2.2 (2.0 to 2.3) | 2.3 (2.2 to 2.5) | 2.4 (2.4 to 2.6) |
| 5 | 3353 (32.1) | 1817 (17.1) | 1134 (10.9) | 4195 (39.9) | 2.2 (2.1 to 2.4) | 2.4 (2.3 to 2.6) | 2.6 (2.4 to 2.7) |

*Mutually adjusted for age, gender, ethnicity and deprivation.
†Mutually adjusted as above, also adjusted for number of long-term conditions (QOF).
NFN, Predicted number of fit notes; QOF, Quality and Outcomes Framework.

by musculoskeletal (21%), infection (17%) and mental illness (14%).

Table 1 shows that the number of fit notes received varied by age, ethnicity and deprivation, but were similar among women and men. Older age, high deprivation, Black Caribbean, Black Other and Mixed ethnic groups were strongly associated with a higher number of fit notes. Twice as many individuals in the oldest age group (51–60 years) received 4+ fit notes compared with those aged 20–25 years. The percentage of individuals receiving 4+ fit notes was higher among Black Caribbean and Black Other individuals (47% and 45%, compared with 34% or less among Other, Asian and White individuals), and for individuals living in more deprived areas (compared with those living in less deprived areas). These differences remained after adjustment for long-term conditions.

Figure 1, table 2 show the number of fit notes by condition at first fit note. Mental illness, stress and musculoskeletal conditions were associated with the highest number of fit notes, after adjusting for demographic factors and long-term conditions. Infection and surgery were associated with the lowest numbers of fit notes. After adjustment for demographic factors and long-term conditions, of the 37 subgroups, the highest predicted number of fit notes were for drug and/or alcohol misuse, common mental disorders and SMI (table 2, online supplemental table 1 and figure 2).

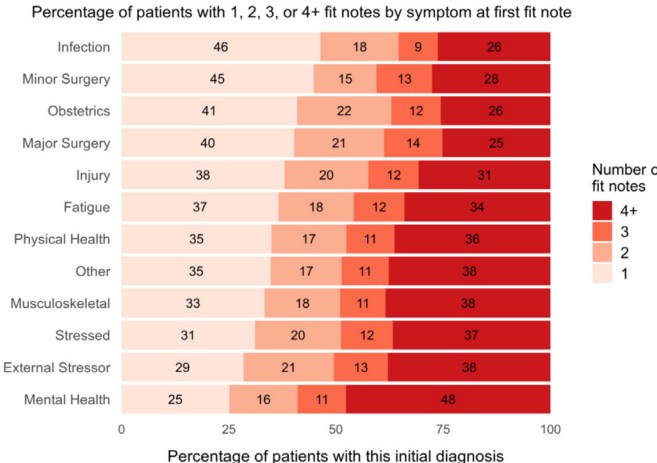

Percentage of patients with 1, 2, 3, or 4+ fit notes by symptom at first fit note

| | | | | |
|---|---|---|---|---|
| Infection | 46 | 18 | 9 | 26 |
| Minor Surgery | 45 | 15 | 13 | 28 |
| Obstetrics | 41 | 22 | 12 | 26 |
| Major Surgery | 40 | 21 | 14 | 25 |
| Injury | 38 | 20 | 12 | 31 |
| Fatigue | 37 | 18 | 12 | 34 |
| Physical Health | 35 | 17 | 11 | 36 |
| Other | 35 | 17 | 11 | 38 |
| Musculoskeletal | 33 | 18 | 11 | 38 |
| Stressed | 31 | 20 | 12 | 37 |
| External Stressor | 29 | 21 | 13 | 38 |
| Mental Health | 25 | 16 | 11 | 48 |

Number of fit notes: 4+, 3, 2, 1

Percentage of patients with this initial diagnosis

Data source: Lambeth Datanet. Sample size: 40698 patients with fit notes aged 16-60, registered with a Lambeth GP between 1st Jan 2014 and 30th April 2017.

**Figure 1** Percentage of patients with 1, 2, 3 or 4+ fit notes by condition at first fit note.

Patients with comorbid mental and physical health conditions had a higher predicted number of fit notes than patients with either mental health or physical health conditions alone. The predicted number of fit notes increased across all condition groups when coupled with a mental health condition, but interaction effects reached statistical significance only for mental health conditions comorbid with infection at first fit note (predicted fit notes for infection 1.8 [1.7,1.8] versus comorbid infection and mental health condition 3.2 [2.4,4.0] ($\chi^2$ 3.7 (1) p=0.05), these results are in the text only.

We found no associations between number of condition types and number of fit notes, but we found a higher number of fit notes among individuals with mental health conditions than those with non-mental health conditions, independent of the number of condition types recorded at first fit note.

## DISCUSSION

We show for the first time that individuals with drug and/or alcohol misuse have the highest number of fit notes of any condition at first fit note. High numbers of fit note represent complexity, chronicity, high service use, longer periods of sickness absence and more frequent primary care reviews. We are aware of one study identifying alcohol and drug misuse recorded on fit notes which found that alcohol and drug misuse was more common in the most socially deprived areas, making up 9% of fit notes for mild–moderate mental health disorders, compared with 1% in the least deprived areas.[35] We anticipate that such analyses of drug and/or alcohol misuse recorded on fit notes (rather than in clinical records as in this study) are likely to lead to an underestimation, because GPs writing fit notes are aware that if they record substance misuse

| Table 2 | Predicted number of fit notes by diagnostic group | | | | | |
|---|---|---|---|---|---|---|
| Category | Groups | Selected subgroups | N people (%) | Predicted number of fit notes (NFN) (95% CI) | Predicted NFN adj* (95% CI) | Predicted NFN adj† (95% CI) |
| Mental health | Total | | 40698 | 2.0 (1.8 to 2.1) | 2.2 (2.1 to 2.3) | 2.4 (2.3 to 2.5) |
| | Mental illness‡ | | 5841 (14.4) | 2.8 (2.6 to 3.0) | 3.3 (3.1 to 3.5) | 3.3 (3.1 to 3.4) |
| | | Drug and alcohol | 584 (1.4) | 3.9 (3.6 to 4.3) | 4.2 (3.8 to 4.6) | 4.5 (4.1 to 4.8) |
| | | Common mental disorders | 4393 (10.8) | 2.7 (2.5 to 2.9) | 3.2 (3.1 to 3.4) | 3.2 (3.1 to 3.3) |
| | | Severe mental illness | 374 (0.9) | 2.7 (2.5 to 3.0) | 3.0 (2.8 to 3.3) | 2.6 (2.4 to 2.8) |
| | Stress | | 1037 (2.6) | 2.1 (1.8 to 2.3) | 2.5 (2.2 to 2.7) | 2.7 (2.4 to 2.9) |
| | External stressor | | 845 (2.1) | 2.2 (1.9 to 2.4) | 2.3 (2.1 to 2.5) | 2.5 (2.3 to 2.7) |
| Non-mental health | MSK | | 8366 (20.6) | 2.2 (2.1 to 2.4) | 2.3 (2.2 to 2.5) | 2.6 (2.4 to 2.7) |
| | Fatigue | | 541 (1.3) | 1.9 (1.6 to 2.2) | 2.2 (2.0 to 2.5) | 2.4 (2.1 to 2.7) |
| | Injury | | 2183 (5.4) | 1.8 (1.6 to 1.9) | 2.1 (1.9 to 2.2) | 2.3 (2.2 to 2.5) |
| | Physical health | | 12710 (31.2) | 2.1 (1.9 to 2.3) | 2.2 (2.1 to 2.3) | 2.3 (2.2 to 2.4) |
| | Other | | 448 (1.1) | 2.0 (1.8 to 2.3) | 2.2 (2.0 to 2.4) | 2.3 (2.1 to 2.5) |
| | Obstetrics | | 1372 (3.4) | 1.4 (1.3 to 1.6) | 2.0 (1.8 to 2.2) | 2.2 (2.0 to 2.4) |
| | Infection | | 6726 (16.5) | 1.5 (1.3 to 1.6) | 1.8 (1.7 to 1.9) | 1.9 (1.8 to 2.0) |
| | Surgery | | 629 (1.6) | 1.5 (1.3 to 1.8) | 1.6 (1.4 to 1.9) | 1.8 (1.5 to 2.1) |

*Adjusted for age, gender, ethnicity and deprivation.
†Adjusted for age, gender, ethnicity, deprivation and number of long-term conditions.
‡Made up of drug and alcohol, common mental disorders, severe mental illness, mental health treatment and mental health other.
MSK, Musculoskeletal condition.

disorders directly on the fit note, it will be seen by the individual's employer.[16]

Across the 11 main groups, our findings were consistent with previous studies; mental illness at first fit note was associated with the highest number of fit notes, followed by 'feeling stressed' and 'external stressors'.[11 36] In contrast, infection and minor surgery were associated with the smallest number of fit notes.[8] Evidence suggests that coexisting mental and physical conditions are associated with poorer clinical and functional outcomes and lower quality of life, than physical conditions alone.[37–48] We found that the combination of mental and physical disorders at first fit note increased the predicted number of fit notes significantly compared with physical conditions alone, but we did not find a 'multiplier effect' of mental health across physical conditions. However, among people who presented with infection and mental health conditions at first fit note, the combination doubled the predicted number of fit notes of infection (p=0.05).

Older individuals, people living in deprived areas and those in Black Other, Black Caribbean and Mixed groups continue to receive on average a higher number of fit notes in the final fully adjusted model. This remaining association is likely to be explained by exposure to a multitude of disadvantages that we have not been able to measure, such as long-term conditions not captured by the QOF, occupational disadvantages such as precarity, high risk occupations (eg, construction, health and social care)[19] and work place discrimination.[49] Although women are more likely to receive a first fit note, we find that they then received a similar number of fit notes to men.

The largest study of fit notes to date is national reporting on fit notes issued, published by NHS Digital.[7] At first glance, compared with NHS Digital, there appears to be an underrepresentation of mental health in our sample, despite being a population with a high prevalence of mental health problems.[50] The lower prevalence of fit notes for mental health in our sample compared with previous studies is expected because we have extracted diagnosis at first fit note, rather than diagnosis on all fit notes, of which mental health takes up a larger proportion due to the association between mental health conditions and multiple fit note use.[14] Second, we know that 1.7%–9.6% of non-mental health diagnosis will change to mental health conditions on future fit notes, which we have not captured in our study.[8]

In contrast to mental health conditions, infections make up a much higher proportion of first fit notes in our sample than total fit note samples (16.5% at first fit note vs 3.5% of NHS Digital sample). Infection is associated with lower number of fit notes and younger populations, so we expect to see a higher proportion of fit notes for infection at first fit note than found in the NHS Digital sample which includes diagnosis on all fit notes, not just the first. In addition, the categorisation of condition types in our study is likely to explain some of the difference: we have categorised conditions using a hierarchy of condition groups, infections were grouped together instead of added to their corresponding system group.[28] Our findings are in keeping with the labour force survey in which 'minor illnesses', including infections, were a large contributor to sickness absence.[51]

The 'drug and/or alcohol misuse' subgroup in our study represents a wide spectrum of conditions, including harmful use and dependence on multiple different substances (online supplemental table 2). Substance misuse, in particular alcohol misuse, is common and associated with absenteeism,[52] and higher risk drinking profiles in early adulthood are an important predictor of receiving a disability pension later in life.[53] The Alcohol Use Disorders Identification Test (AUDIT) used in the Adult Psychiatric Morbidity Survey 2014 found that 20% of adults in England drink at hazardous levels and above (AUDIT ≥8). White men and women are more likely to drink at hazardous, and harmful levels than Black, Asian and minority ethnic groups. Some workforce groups, such as doctors, and male-dominated industries, such as the military and construction, have been identified as having particularly high levels of alcohol misuse.[54–56] Despite high levels of physical and mental comorbidity, workers with drug and/or alcohol misuse often present late, and struggle to access health services.[57] In addition, in recent years funding for addiction services has been drastically cut in England, and addiction budgets removed from ringfencing protections within the NHS.[58 59] The findings suggest a case for employment based treatment services such as the successful in-house Drug and Alcohol Treatment Service developed by Transport for London,[60] a major employer in Lambeth.

Patients with multiple fit notes are groups with high service use, chronicity and complexity, who may benefit from earlier intervention and focused occupational health support. We found that communities with highest number of fit note use are at highest risk of insecure work, and workers with mental health conditions, particularly drug and/or alcohol misuse, experience major barriers to work. Countries in which changes have been most successful in tackling exclusion from the workplace due to sickness and disability, include those with systems for early identification of occupational health problems and where employer incentives and legislation have supported workers to exercise their right to remain in work.[61] Further research is needed into access to healthcare and occupational support among patients with multiple fit note use.

This is the largest study to analyse fit note use and clinical information from GP records at point of first fit note, using individual-level clinical data. A major strength of using clinical records data is that they contain more detail on condition type than information reported on the fit note which previous research has relied on. Clinical record data also provided usable information on reason for fit note for 98% of first fit notes, compared with the 50% of NHS Digital fit notes with diagnostic information. The study was limited because we do not have access to the information written on the fit note to compare it with

the information recorded in clinical notes, and we do not know the length of time that each fit note was prescribed for, or the number of sickness absence episodes. Therefore, we are unable to describe the precise pattern of multiple fit note use. For example, higher numbers of fit notes could be due to longer periods of sickness absence, or more frequent reviews in response to diagnostic uncertainty, negotiation of sickness absence or provision of health care.[62] We cannot differentiate whether a high predicted number of fit notes represent longer sickness absence, a tendency for more frequent review by GPs, or both. We also did not have access to sickness certification before the start of our study window; therefore, our results do not represent lifetime first fit note, but first fit note in the study window. Each of the condition groups and subgroups encompasses a very wide variety of disorders. For example, the subgroup 'drug and/or alcohol misuse' is made up of harmful use and dependence diagnoses on multiple different substances (see online supplemental table 2). The lack of granularity in the 'drug and/or alcohol' subgroup, and other groups/subgroups is a limitation.

The study was conducted in a single geographical area for which we had GP data. This had the benefit of providing information on virtually all individuals seeking help within that population, but it had the disadvantage of a possible loss of generalisability, and our findings should be tested in other areas. There is additional information that would have been extremely useful, such as nature of employment, educational level or benefit use, which were not available in this data set. Without this information, it was impossible to ascertain whether fit notes were issued for individuals in work versus those already out of work who were applying for health benefits. Furthermore, we did not know the occupational status of individuals within the population—for example, whether they were unemployed, retired or in full-time education.

## CONCLUSIONS

For the first time, we show drug and/or alcohol misuse at first fit note is associated with the highest number of fit notes, and that there is evidence that comorbid mental health conditions at first fit note may have a multiplier effect on predicted number of fit notes for individuals presenting with infection. Demographic risk factors for high numbers of fit notes include Black Caribbean, Black Other or Mixed ethnicity, and higher levels of deprivation; these are likely to be explained by risk factors that we have not been able to measure, such as high risk occupations[19] and work place discrimination.[49] An understanding of the predicted number of fit notes by condition and demographic group enables policy-makers to evaluate the primary care resources available to working age adults at high risk of multiple fit note receipt. Linked occupational and health data sets are needed to better understand the occupational, clinical and wider structural risk factors for

high number of fit notes, and to explore the trajectories of individuals after first fit note use.

**Acknowledgements** Particular thanks to Megan Pritchard and Amelia Jewell at the NIHR Maudsley Biomedical Research Centre for their support with this study.

**Contributors** SD designed the study, analysed the data and drafted the manuscript. MH, SH and IM advised extensively throughout the process. EC advised on the statistical analysis. CP advised on the categorisation of read codes into groups and subgroups. MB, MA and SS provided expertise and guidance on extracting, analysing and interpreting the data. ER and all authors commented on the final manuscript.

**Funding** This paper represents independent research part funded by the Royal College of Psychiatrists' Donald Dean Fellowship and the National Institute for Health Research (NIHR) Biomedical Research Centre at South London and Maudsley NHS Foundation Trust and King's College London (IS-BRC-1215-20018). The views expressed are those of the authors and not necessarily those of the Royal College of Psychiatrists, the NHS, the NIHR or the Department of Health and NIHR Maudsley Biomedical Research Centre, South London and Maudsley NHS Foundation Trust, London, UK.

**Competing interests** Professor MH receives funding from Janssen as part of the RADAR-CNS consortium, which includes a project on depression. He is a principal investigator of RADAR-CNS, a precompetitive public private partnership cofunded by Innovative Medicines Initiative (European Commission) and European Federation of Pharmaceutical Industries and Associations (EFPIA). He has also been an independent expert witness in group litigations instructed by claimants against pharmaceutical companies for alleged harmful effects of their products. Authors have no other conflict of interest to declare.

**Patient and public involvement statement** Findings from this study were presented to and discussed with the NIHR Biomedical Research Nucleus Data Linkage Service User and Carer Advisory Group.

**Patient consent for publication** Not required.

**Ethics approval** CRIS was established in 2008 and approved by the Oxfordshire Research Ethics Committee in 2008 (reference 18/SC/0372). Approval for linkage with Lambeth DataNet was granted by Lambeth Clinical Commissioning Group and Information Governance Committee. This project was approved by the CRIS Oversight Committee in 2015. All patients have the choice to opt-out of their anonymised data being used.

**Provenance and peer review** Not commissioned; externally peer reviewed.

**Data availability statement** Data may be obtained from a third party and are not publicly available. Access to the deidentified data in this study can be applied for through Lambeth Datanet and South London and Maudsley Biomedical Research Centre (BRC), De Crespigny Park, London SE5 8AF (cris.administrator@slam.nhs.uk). Access is granted only if approval is given by both the BRC and the Lambeth Datanet Steering Group.

**ORCID iDs**
Sarah Dorrington http://orcid.org/0000-0002-6462-1880
Ewan Carr http://orcid.org/0000-0002-1146-4922
C Polling http://orcid.org/0000-0003-2657-0696
Mark Ashworth http://orcid.org/0000-0001-6514-9904
Emmett Roberts http://orcid.org/0000-0002-4152-5570

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
