## [Reviewer comments · BMJ Open]

ARTICLE DETAILS

TITLE (PROVISIONAL)	Health condition at first fit note and number of fit notes: a longitudinal study of primary care records in south London
AUTHORS	Dorrington, Sarah; Carr, Ewan; Polling, C; Stevelink, Sharon; Ashworth, Mark; Roberts, Emmert; Broadbent, Matthew; HATCH, STEPHANI; MADAN, IRA; Hotopf, Matthew

VERSION 1 – REVIEW

REVIEWER	Mark Gabbay University of Liverpool
REVIEW RETURNED	23-Sep-2020

GENERAL COMMENTS	This is an interesting paper. The main problem with it is that it isn't as clear as it could be about the 'so what'. As I read the paper, it sets out the data on the likelihood of repeat fitnotes for an episode of sickness absence by condition. However it may be that this represents longer absence, more uncertainty about prognosis, or a tendency for shorter notes and more regular review. It is a pity that the researchers were unable to download electronic fitnotes for these cases to ascertain the diagnosis given on the note to compare it with the key diagnoses at the index consultations, or the length of the note or length of the total period of absence. It's not clear what the primary outcome being measured is, but I think it is total number of notes for same absence period (continuous presumably) after first note issued within the timeframe of the study. Of course presumably some of these may not be the first note in a series if appearing for the first time in the early part of the sampling period, as presumably there is no way to check this by the data collection method used. Furthermore other research suggests that for some conditions a shorter note is offered as part of the negotiation and request for review either of absence or the ongoing care (see for example Byrne P, Ring A, Salmon P, Dowrick C, Gabbay M Tussles and rollovers: negotiating sickness absence in primary care Advances in Applied Sociology; 2014; 4:247-260 DOI: 10.4236/aasoci.2014.412029). It may also represent therefore more close follow-up for conditions where more support is required or more supervision of the requirement of a note or uncertainty of about return to work or work modifications etc. It may also be an uncertainty about whether the patient should be off or poor continuity and thus less confidence about commitment to a predicted period of absence. It was expected that the fitnote would promote more regular reviews and a push towards earlier return to work and workplace modification, phased returns etc so this increased number of notes may represent conditions where this is commoner? All of these issues I suggest need addressing in the discussion and the primary question clarifying- number of notes for single absence episode and also justifying in the introduction, and returning to in the discussion- why does it matter how many notes there are for an episode of absence- what behaviours might this represent about certification, and significance wrt absence length, chronicity and risk of prolonged absence.
---

	Finally the issue about the lack of link to the electronic fitnote data is a problem that needs acknowledging and exploring in more depth, as do other factors such as whether the patients are employed or on welfare benefits, practitioner characteristics, gender age deprivation. It's not clear to me on reading the paper what the association is between these factors and principal diagnosis. The authors need to be clear about how it was determined which condition to classify the diagnosis as when presumably in some cases at least, multiple conditions were recorded at the index consultation. What is surprising perhaps as shown in figures 1 and 2 is that there were at least 25% of patients in all categories that had at least 4 notes, and I fear some of these data are insufficiently explored as to what might be going on in these consultations and fitnote discussions and decisions. Finally reference 9 in the reference list seems to be different from that cited unless the Dorrington in press is another paper not appearing in the list (which surely it should be if it's accepted and in press)
--	---

REVIEWER	Dr Kevan Thorley Devon and Cornwall Hath Care Trust UK
REVIEW RETURNED	29-Sep-2020

GENERAL COMMENTS	This is a report of an important, large scale study which provides useful information for policy development. The method of starting from first fit note issue and exploring GP records for diagnoses and associated factors is novel and provides insights which have not been previously available. Two minor revisions would render the report more readily understandable to the generalist reader  1. The aim of the study could be more simply stated. On page 4 at the end of the introduction there is a paragraph which include method and objectives. Some disentanglement of these would improve clarity. 2. It would be helpful to explain in greater detail how the predicted number of fit notes is derived, again to make the report more readily understood by the generalist reader but also to strengthen th conclusions. Some comment about future comparison of rpedicted and actual number of fit notes issued might be appropriate.
--

VERSION 1 – AUTHOR RESPONSE

Reviewer comments	Authors response
1 This is an interesting paper. The main problem with it is that it isn't as clear as it could be about the 'so what'. As I read the paper, it sets out the data on the likelihood of repeat fitnotes for an episode of sickness absence by condition.	Thank you for raising this important point, we have clarified the 'so what' factor throughout the paper. Introduction: "A single fit note following an injury or illness is a common occurrence. A study found that 60% of all fit notes issued in the UK from 2011-13 were single fit notes⁸. In contrast, repeated fit notes are rarer and suggest a more serious injury or chronic illness, or other reasons for not recovering, such as multimorbidity, illness complications or barriers to return to work. Identifying conditions and demographic groups associated with high fit note use enables us to define groups with high levels of need and to inform future interventions." Discussion: "High numbers of fit note represent a combination of complexity, chronicity, high service use, longer periods of sickness absence and more frequent primary care reviews." "Patients with multiple fit notes are groups with high service use, chronicity and complexity, who may benefit from earlier intervention and focused occupational health support. Further research is needed into access to healthcare and occupational support amongst patients with multiple fit note use." Conclusion "An understanding of the predicted number of fit notes by condition and demographic group enables policy makers to evaluate the primary care resources available to working age adults' at high risk of multiple fit note receipt."
However it may be that this represents longer absence, more uncertainty about prognosis, or a tendency for shorter notes and more regular review.	We agree there are major limitations to working with primary care health records. The strengths of health records are in describing patterns of health care use in large, inclusive, diverse populations. These strengths come with the frustrating limitations of health records, which you describe throughout your review. This has enabled us to clarify the limitations throughout our paper. Health records describe patterns which form a starting point for the development of a more in-depth analysis; for example, of the experience of patients who require multiple fit notes, and the GPs providing them. We describe the limitations of health records throughout the paper. In addition we have added the important points you raise to the limitations section: "The study was limited because we do not have access to the information written on the fit note to compare it with the information recorded in clinical notes, and we do not know the length of time that each fit note was prescribed for. Therefore we are unable to describe the precise pattern of multiple fit note use. For example, higher numbers of fit notes could be due to longer periods of

		sickness absence, or more frequent reviews in response to diagnostic uncertainty, negotiation of sickness absence or provision of health care⁶². We cannot differentiate whether a high predicted number of fit notes represent longer sickness absence, a tendency for more frequent review by GPs, or both.
	It is a pity that the researchers were unable to download electronic fitnotes for these cases to ascertain the diagnosis given on the note to compare it with the key diagnoses at the index consultations, or the length of the note or length of the total period of absence.	We agree. This is a limitation described in the summary limitations section, introduction and discussion.
	It's not clear what the primary outcome being measured is, but I think it is total number of notes for same absence period (continuous presumably) after first note issued within the timeframe of the study.	Thank you for this opportunity to clarify, the primary outcome is total number of fit notes during the period Jan 2014-May 2016. We have clarified this in the abstract and introduction. Abstract: “The fit note replaced the sick note in the UK in 2010, with the aim of improving support for patients requiring sickness absence, yet there has been very little research into fit note use. This study aims to describe number of fit notes by condition, to improve our understanding of patterns of fit note use in primary care.” “Primary outcome measure: Predicted number of fit notes in the period Jan 2014-May 2016” Introduction “We explore the relationship between variables known to be associated with fit note receipt and number of fit notes, anticipating that the number of fit notes issued in our sample, between Jan 2014-April 2017, would be higher among individuals with mental disorders, those with multimorbidity, women, those living in areas of greater deprivation, older age groups, and minority ethnic groups. We will test for a multiplier effect of mental health and physical co-morbidity on number of fit notes. We expect demographic associations to be partially explained by long-term conditions and that individuals with infection, injury and obstetric conditions will tend to have a fewer fit notes.”

	Of course presumably some of these may not be the first note in a series if appearing for the first time in the early part of the sampling period, as presumably there is no way to check this by the data collection method used.	You are correct, we have not limited this to a single absence period. This is clarified in the limitations section. “The study was limited because we do not have access to the information written on the fit note to compare it with the information recorded in clinical notes, and we do not know the length of time that each fit note was prescribed for, or the number of sickness absence episodes.”
	Furthermore other research suggests that for some conditions a shorter note is offered as part of the negotiation and request for review either of absence or the ongoing care (see for example Byrne P, Ring A, Salmon P, Dowrick C, Gabbay M Tussles and rollovers: negotiating sickness absence in primary care Advances in Applied Sociology; 2014; 4:247-260 DOI: 10.4236/aasoci.2014.412029). It may also represent therefore more close follow-up for conditions where more support is required or more supervision of the requirement of a note or uncertainty of about return to work or work modifications etc. It may also be an uncertainty about whether the patient should be off or poor continuity and thus less confidence about commitment to a predicted period of absence.	Thank you these are important points that we have added to the discussion. Thank you for the citation, we have also included this in the discussion. We hope to apply mixed methods in future work to explore these questions fully. Discussion (limitations) “The study was limited because we do not have access to the information written on the fit note to compare it with the information recorded in clinical notes, and we do not know the length of time that each fit note was prescribed for. Therefor we are unable to describe the precise pattern of multiple fit note use. For example, higher numbers of fit notes could be due to longer periods of sickness absence, or more frequent reviews in response to diagnostic uncertainty, negotiation of sickness absence or provision of health care⁶². We cannot differentiate whether a high predicted number of fit notes represent longer sickness absence, a tendency for more frequent review by GPs, or both.”
	It was expected that the fitnote would promote more regular reviews and a push towards earlier return to work and workplace modification, phased returns etc so this increased number of notes may represent conditions where this is commoner?	Thank you. We have added this point to the introduction, although we cannot address this question in this study, which we have clarified in the limitations section of the discussion. The fit note was designed to reduce long term sickness absence by replacing the sick note’s binary ‘fit’ versus ‘not fit to work’, with the addition of a third option, ‘maybe fit’², and encouraging GPs to certify shorter periods of sickness absence, with more frequent reviews.
	All of these issues I suggest need addressing in the discussion and the primary question clarifying- number of notes for single absence episode and also justifying in the introduction, and returning to in	Thank you the points have been added to the discussion, and the primary question has been clarified throughout the paper (as already described above in response to your first point): Introduction: “A single fit note following an injury or illness is a common occurrence. A study found that 60% of all fit notes issued in the UK

	the discussion- why does it matter how many notes there are for an episode of absence- what behaviours might this represent about certification, and significance wrt absence length, chronicity and risk of prolonged absence.	from 2011-13 were single fit notes⁸. In contrast, repeated fit notes are rarer and suggest a more serious injury or chronic illness, or other reasons for not recovering, such as multimorbidity, illness complications or barriers to return to work. Identifying conditions and demographic groups with high fit note use enables us to define groups with high levels of need who could form the focus of future occupational health interventions. Discussion: “High numbers of fit note represent a combination of complexity, chronicity, high service use, longer periods of sickness absence and more frequent primary care reviews.” “Patients with multiple fit notes are groups with high service use, chronicity and complexity, who may benefit from earlier intervention and focused occupational health support. Further research is needed into access to healthcare and occupational support amongst patients with multiple fit note use.” Conclusion “An understanding of the predicted number of fit notes by condition and demographic group enables policy makers to evaluate the primary care resources available to working age adults’ at high risk of multiple fit note receipt.”
	Finally the issue about the lack of link to the electronic fitnote data is a problem that needs acknowledging and exploring in more depth, as do other factors such as whether the patients are employed or on welfare benefits, practitioner characteristics, gender age deprivation. It's not clear to me on reading the paper what the association is between these factors and principal diagnosis.	We hope we have understood your question correctly. Thank you for the opportunity to clarify this. Brief limitations section:  • “We do not have access to the information written on the fit note to compare it with the information recorded in clinical notes, and we do not know the length of time that each fit note was prescribed for.” • “There is additional information that would have been extremely useful, such as nature of employment, educational level, occupational status or benefit use, which were not available in this dataset.” Within the data we have a linkage which enables us to access some essential electronic fit note data but due to ethical restrictions we are unable to access written text on fit notes. The unavailable data includes the diagnosis written on the fit note, and the length the fit note is prescribed for. This is stated in the summary, the introduction and the limitations section. Instead we worked with clinical data recorded at the time of first fit note, this provided us with more information than would have been available on the fit note (which is missing or not clearly stated in around 50% of fit notes). This is described in the paper. You are correct, we do have information on gender, age and deprivation, and we include these in our analysis model. There are well known associations between demographic groups and different conditions but as this is not the focus of the paper we do not describe these associations in this paper. We don't have information on employment or welfare benefits, therefore we are unable to define associations between employment/welfare and conditions. This is summarised in the summary limitations section. We suggest in the paper that

		demographic variation may be partly explained by demographic variation in employment and welfare. This is described in the discussion section of the paper. Unlike previous studies of fit note use, we were unable to analyse practitioner characteristics in this study. Instead, practice variation was controlled for using clustering. We hope to address the questions raised in future work with data sets and methods that allow for a deep dive into the trajectories of patients.
	The authors need to be clear about how it was determined which condition to classify the diagnosis as when presumably in some cases at least, multiple conditions were recorded at the index consultation.	86% of people had one condition at first fit note, 12% had two conditions and 2% had three conditions. When there was more than one condition at first fit note the primary condition was selected at random. The other conditions were taken into account in the analysis of comorbid conditions. Methods: “For patients with more than one condition at first fit note (14% of people) one condition was selected at random as the primary condition, additional conditions were taken into account in the analysis of multiple symptoms.”
	What is surprising perhaps as shown in figures 1 and 2 is that there were at least 25% of patients in all categories that had at least 4 notes, and I fear some of these data are insufficiently explored as to what might be going on in these consultations and fitnote discussions and decisions.	Thank you, this is a very important point. The group with over 4 fit notes are of particular interest and will be explored in more detail in future mixed methods work. Using health records we are unable to access more detailed information on fit note discussion and decisions.
	Finally reference 9 in the reference list seems to be different from that cited unless the Dorrington in press is another paper not appearing in the list (which surely it should be if it's accepted and in press)	Thank you, this paper has now been published and the reference has been updated.
2	This is a report of an important, large scale study which provides useful information for policy development. The method of starting from first fit note issue and exploring GP records for diagnoses and associated factors is novel and provides insights which have not been previously available. Two minor revisions would render the report more readily understandable to the generalist reader	Thank you for your comments.
	1. The aim of the study could be more simply stated. On page 4 at the end of the introduction there is a paragraph which	Thank you, this is very helpful. We have disentangled the method and objectives as you have suggested, and revised the study aim so that it is more simple stated.

	include method and objectives. Some disentanglement of these would improve clarity.	For clarity we have also summarised the aims of the paper: “In summary, the first aim of this paper is to describe the predicted number of fit notes by condition at first fit note extracted from clinical notes. The second aim is to describe demographic variation in predicted number of fit notes, lastly we will explore interaction effects of co-morbid conditions at first fit note on predicted number of fit notes.”
	2. It would be helpful to explain in greater detail how the predicted number of fit notes is derived, again to make the report more readily understood by the generalist reader but also to strengthen the conclusions. Some comment about future comparison of predicted and actual number of fit notes issued might be appropriate.	Thank you for the opportunity to clarify this. The text in bold has been added to the methods for clarity. The predicted number of fit notes were derived from the negative binomial model using the "margins" function in Stata. This provides a model-derived prediction of the number of fit notes received for each group or subgroup, accounting for other variables in the model (e.g., age, gender) and the clustering of individuals within GP practices. In our case, we calculate the predicted counts per group holding other variables to their mean values. “Regarding the difference between predicted and actual counts: these are connected but different. The actual counts are the counts observed in the raw data, without any adjustment or clustering. The predicted counts are derived from the regression model, and as such, account for other variables in the model.” We think it is important to account for variables such as age, gender, ethnicity and deprivation, that have previously been associated with health status and fit note use. The two estimates -- actual counts and model-derived predicted counts -- are therefore comparable and closely connected. If the included covariates were uncorrelated with the outcome, we would expect the two counts (actual and predicted) to be identical. However, as we account for important variables associated with the outcome, the two counts will deviate. The 'actual' count will then represent the collected data; the 'predicted' count will represent the number of fit notes after accounting for the included covariates. Both numbers are of interest in interpreting these results.